# OpenReview forum: "RESTOR: Knowledge Recovery in Machine Unlearning"
_TMLR — Accepted by TMLR_

### Review · Reviewer_kKUp · 2025-03-12

**Summary Of Contributions:**

This paper studies machine unlearning algorithms and evaluation methods from the perspective of how much of the actual full effect of erasing a datapoint can be achieved with unlearning (i.e., being identical to "retraining"), by assessing successful knowledge recovery performance. In this context, authors introduce the concept of "restorative unlearning" as knowledge-oriented framework, which does not only aim to forget the desired content but also aim to restore the model to its original knowledge state. The paper specifically focuses on natural language processing applications, and forgetting of manipulated content in documents. Several existing approximate machine unlearning method baselines are examined in this setting, based on customized factual and corruption datasets.

**Audience:**

Yes

**Claims And Evidence:**

Yes

**Requested Changes:**

- The research community would significantly benefit from the material accessibility of this work, particularly the well designed factual and corruption datasets that are constructed for this paper. Perhaps the paper could include a reproducibility statement regarding this, by referring to the content in the supplementary.

- Assessing a model's knowledge before any corruption in practical, larger scale applications/datasets to evaluate unlearning algorithms would potentially be computationally challenging (or infeasible)? I would suggest perhaps a discussion on this to be included in the paper.

- Minor typo in Table 1 caption: "Gradient Ascent".

**Strengths And Weaknesses:**

Strengths:
- The scope of the paper is very interesting and the introduced concept of restorative unlearning is a novel perspective. This is particularly important in the context of unlearning for language models.
- The paper is very well written and it is of high quality experimentally as well.
- Empirical results provide very interesting insights regarding the evaluations of some existing approximate unlearning baselines (Sec 4.3 broadly).

Weaknesses: I do not see any significant weaknesses that would contradict the support of the claims and evidence provided in the paper.

---

> ### Author Response · Authors · 2025-03-13
>
> We appreciate the reviewer’s detailed and positive feedback. We are glad that you find our work novel and interesting and that you recognize the quality of our experiments and writing. Regarding the requested changes:
>
> > The research community would significantly benefit from the material accessibility of this work, particularly the well-designed factual and corruption datasets constructed for this paper. Perhaps the paper could include a reproducibility statement regarding this by referring to the content in the supplementary.
>
> Thank you for the suggestion. We will ensure that both datasets, along with the procedure to systematically generate them, are made available and easy to use.
>
>
> > Minor typo in Table 1 caption: “Gradient Ascent”.
>
> We appreciate you catching this—we will correct the typo in the revised version.
>
>
> > Assessing a model’s knowledge before any corruption in practical, larger-scale applications/datasets to evaluate unlearning algorithms would potentially be computationally challenging (or infeasible)? I would suggest perhaps a discussion on this to be included in the paper.
>
> We will include a discussion on this point in the paper and agree that further research on restorative unlearning is needed for larger-scale applications. Our work serves as a stepping stone for restorative unlearning, with our primary focus on establishing a framework to evaluate this new perspective. To that end, we designed a knowledge-oriented task where systematic evaluation remains feasible.

---

### Review · Reviewer_BuHh · 2025-03-17

**Summary Of Contributions:**

Restore provides a new perspective on machine unlearning for LLMs by going beyond forgetting and investigating the ability of unlearning methods to restore knowledge that has been corrupted in LLMs.

**Audience:**

Yes

**Claims And Evidence:**

Yes

**Requested Changes:**

**(A) Prior literature**

While restorative unlearning is new in the LLM space, prior works have investigated this space and given it different names. E.g., [1] "Corrective unlearning" which was further specified down to "healing" in [2]

Excerpts:
[1]: A corrective unlearning algorithm Ucorr “corrects” the original model Mo by removing the influence of Sm. (not that close)

[2]: We will refer to **going beyond forgetting** the poison trigger and **also causing the model to perform a correct classification** as **healing** poisoned data. (very close to the Restore definition)

---

[1] Goel, Shashwat, et al. "Corrective Machine Unlearning." Transactions on Machine Learning Research (2024).

[2] Schoepf, Stefan, Jack Foster, and Alexandra Brintrup. "Potion: Towards Poison Unlearning." Journal of Data-centric Machine Learning Research (2024).

---
**(B) Privacy aspect of unlearning** (feel free to ignore)

How do the MIA scores change for samples that were able to be restored versus those that point to wrong answers? Is restoring an indication of ideal unlearning in the privacy sense?

---
**(C) Going beyond factual knowledge** (feel free to ignore)

While not necessary for acceptance (in my opinion), what results can be observed when doing the same experiments on more complex relationships such as belief manipulation? Example: Testing for the bias of a model before corruption (e.g. given x options of products which will it choose) and seeing if unlearning reverts the model back from a corrupted state to the same belief. This could address recent challenges mentioned in [3].

---
[3] Zhang, Yiming, Javier Rando, Ivan Evtimov, Jianfeng Chi, Eric Michael Smith, Nicholas Carlini, Florian Tramèr, and Daphne Ippolito. "Persistent Pre-Training Poisoning of LLMs." arXiv preprint arXiv:2410.13722 (2024).

---
**(D) Judge model choice**

Why did the authors choose GPT-3.5  as a judge given much more capable available models?

---
**(E) 4.3 k=0 further details**

*In fact, it seems that only providing incorrect facts (k = 0), does not effectively
change model’s underlying knowledge over entities, and having a context results in longer, more diverse
samples, and more effective corruption.*

Please further elaborate on this in the paper. It seems like a noteworthy aspect to understand and might have further implications on the experiments and results.

---
**(F) Minor comments**

- Please check sentences for grammatical errors
- Figure 2. is confusing at first, please add a legend title specifying that the colors are for models since you use clean, corrupted, ... for the x-axis and the legend
- Adding the explanation of k in the text of Table 1 would make it easier for readers that skim the tables at their first read

**Strengths And Weaknesses:**

Strengths:

The insights on which methods work for restoring knowledge, how the share of irrelevant information in the forget set impacts restoring quality, and correlations of prior accuracy to restoring quality are strong contributions to the unlearning literature on LLMs. The paper is well-written and I did not see any major problems in the methodology.

Weaknesses: See requested changes
As a note, I do not see any of the weaknesses as big problems, but rather as chances to make the paper even better.

---

> ### Author Response · Authors · 2025-03-27
>
> We appreciate the reviewer for detailed and helpful comments, and we will incorporate all your suggestions. We are glad that you find the insights provided in the paper to be a valuable  contribution to the unlearning literature, and the paper to be well-written. In what follows we address your requested changes.
>
>
> > (A) Prior literature
>
> We appreciate the reviewer’s comments on related works relevant to our definition of restorative unlearning. We will incorporate these into the draft, particularly Schoepf et al., which addresses the removal of poisoned samples to achieve corrective unlearning. However, our analysis is specifically designed for unlearning challenges in the language domain rather than the vision domain.
>
>
> > (C) Going beyond factual knowledge
>
> We appreciate the reviewer’s insightful comment. We too believe that restorative unlearning could be further explored in other scenarios, such as belief manipulation through bias injection and data poisoning, as highlighted in the Conclusion and Limitations (Appendix) sections.
>
> Restorative unlearning is definitely needed for applications like belief manipulations, as the knowledge about entities being manipulated should be restored to the original (unbiased) knowledge. We focus on a knowledge-oriented perspective of restorative unlearning with factual evaluation, as it allows for scalable and systematic assessment. We hope our work serves as a foundation for future studies on restorative unlearning in LLMs, with these other challenges being interesting avenues for further exploration.
>
> > (D) Judge model choice
>
> We chose GPT-3.5 because it was both cost-effective and sufficiently capable for our analysis, i.e., accepting an output if it is relevant enough to the correct answers. As discussed in Appendix G, our experiments showed that GPT-3.5 achieves reasonable accuracy when compared to human evaluation.
>
>
> > (E) 4.3 k=0 further details
>
> We appreciate your detailed comment. Interestingly, we observed that in the pretraining stage, corrupting the model’s knowledge is more effective when incorrect facts are interleaved within a broader context. In this case, when the model is evaluated on these facts, it is more likely to generate incorrect predictions. On the other hand, directly injecting incorrect information and fine-tuning a clean model on these documents—using next-token prediction loss over all tokens as the objective—does not significantly alter the model’s knowledge of the targeted entities, as reflected in the minimal drop in accuracy. This suggests that in pretraining incorrect facts are more persuasive when embedded within broader, seemingly credible contexts, much like propaganda mixing falsehoods with unrelated truthful information. Please see Appendix B for some of the examples.
>
> Note that we are simulating the pretraining stage by fine-tuning over all tokens using cross-entropy loss. Fine-tuning the model to predict an answer given a specific question could potentially exhibit different behaviors. For example, as observed in existing literature like TOFU [1], corruption occurs when the model is fine-tuned to predict an answer given a question, rather than during the pretraining stage. In those setups, incorrect responses are directly used for fine tuning, and the process is effective.
>
> In our experiments, we utilized diverse datasets with varying levels of corruption and applied unlearning algorithms, enabling us to evaluate their effectiveness across different degrees of corruption.
>
> > (F) Minor comments
>
> > Please check sentences for grammatical errors
>
> We will make sure to check this again.
>
> > Figure 2. is confusing at first, please add a legend title specifying that the colors are for models since you use clean, corrupted, ... for the x-axis and the legend
>
> Thanks for pointing this out, we’ll add the legend to better clarify the figure.
>
> > Adding the explanation of k in the text of Table 1 would make it easier for readers who skim the tables at their first read
>
> Thanks for raising this point, we’ll clarify what the value of $k$ means in the main Table.
>
>
>
>
> [1] Maini, Pratyush, et al. "Tofu: A task of fictitious unlearning for llms." arXiv preprint arXiv:2401.06121 (2024).

---

> > ### Comment · Reviewer_BuHh · 2025-03-31
> > **Thank you for your reply**
> >
> > Thank you for your detailed reply, especially on (E). From my perspective this paper fulfills the TMLR criteria.
> >
> > As a side note, it would be incredibly interesting to see how an unlearned model behaves when it is fine tuned multiple times with a shifting target. E.g. the answer of the original model is A. After the first fine tuning it answers with B. After another fine tuning it answers with C. What will it answer with after unlearning now - A or B?
> > Feel free to ignore this for now and see it as interesting future work.

---

> > > ### Author Response · Authors · 2025-04-01
> > >
> > > We appreciate that you find our paper a fit for TMLR.
> > >
> > > > As a side note, it would be incredibly interesting to see how an unlearned model behaves when it is fine tuned multiple times with a shifting target. E.g. the answer of the original model is A. After the first fine tuning it answers with B. After another fine tuning it answers with C. What will it answer with after unlearning now - A or B? Feel free to ignore this for now and see it as interesting future work.
> > >
> > > Thanks for suggesting this idea! It would indeed be interesting to explore how unlearning affects such a sequential scenario. In fact, the order in which documents are unlearned could play a role in this process. We agree that this would be an interesting direction for future study.

---

### Review · Reviewer_ExCS · 2025-03-20

**Summary Of Contributions:**

The paper introduces a new evaluation of machine unlearning in language models which removes the influence of specific corrupted data while preserving the knowledge unrelated to that data. To achieve this, the authors construct a dataset comprising 1051 facts about 50 famous individuals, where the dataset includes varying levels of unrelated context. The paper also studies the performance of existing methods by comparing the performance of a clean model, a corrupted model, and an unlearned model.

**Audience:**

Yes

**Claims And Evidence:**

Yes

**Requested Changes:**

1) The paper should test different numbers of fine-tuning epochs to determine how deeply the corruption affects the model’s knowledge and how that influences unlearning.
2) The hyperparameters of fine-tuning should be examined more thoroughly to understand how they impact both the strength of corruption and the effectiveness of subsequent unlearning.
3) Finally, the paper should clarify whether and how the insights gained in the current experimental setting would generalize to full-scale pretraining scenarios.

**Strengths And Weaknesses:**

Strengths:

1) The evaluation proposed by this paper is interesting and reasonable.

2) The paper is well-written.

Weaknesses:

1) The authors fine-tune a clean model on the corrupted dataset for a fixed number of epochs, but do not provide an in-depth exploration of how varying the number of corruption epochs affects model degradation and unlearning effectiveness. Since fine-tuning schedules can significantly impact a model’s knowledge state, a more thorough investigation or ablation on different epoch counts would strengthen the benchmark’s utility.

2) The paper simulates unlearning by fine-tuning a clean model on a corrupted dataset. In real-world scenarios, large language models are trained on vast corpora that may include sensitive or erroneous data. It remains unclear whether insights from these toy fine-tuning scenarios will fully translate to real LLM pretraining pipelines.

---

> ### Author Response · Authors · 2025-03-27
>
> We appreciate the reviewer’s detailed comments, and the fact that the reviewer finds our study interesting, and the paper well-written.
>
> > The authors fine-tune a clean model on the corrupted dataset for a fixed number of epochs, but do not provide an in-depth exploration of how varying the number of corruption epochs affects model degradation and unlearning effectiveness. Since fine-tuning schedules can significantly impact a model’s knowledge state, a more thorough investigation or ablation on different epoch counts would strengthen the benchmark’s utility.
>
> We appreciate the reviewer’s comment, and ran experiments over the response period to follow the reviewer's suggestion and will describe them below.
>
> It is also worth noting that in our experiments, to achieve different levels of corruption, we considered different datasets (these datasets defined by the value of $k$, which controls the degree of unrelated facts in them) and maintained a consistent corruption procedure across them. As shown in Table 1 (in the draft), we obtain five corrupted models with different corruption levels (measured by the drop in accuracy), and these models are then processed using the unlearning algorithms. Table 1 (in the draft) illustrates how each algorithm performs on each of the corrupted models.
>
> However, during the response period, we followed your suggestion and conducted additional experiments involving corruption scenarios with the same corrupted dataset but varying numbers of corruption epochs to achieve different levels of corruption.
> For this, we used NPO and GradDiff as unlearning algorithms, and here are the results:
>
> | Epochs (e) | Corrupted Model | GA | NPO   |
> |------------|----------------|----------|-------|
> | 3          | 62.15          | 36.16    | 61.99 |
> | 5          | 58.37          | 38.00    | 61.01 |
> | 8          | 51.84          | 33.98    | 60.20 |
> | 10         | 50.22          | 34.73    | 58.86 |
>
> As seen above, training for more epochs makes corruption more severe. However, NPO is able to recover effectively regardless of the corruption level, although there is a slight decrease in performance as the level of corruption increases. In contrast, GA is unable to recover, and its performance also declines with higher corruption levels.
> Note that the above results align with the trend observed in Table 1 of the draft, where NPO was able to recover accuracy even as corruption became more severe, albeit with a slight decrease in performance. Once again, we appreciate the suggestion for this experiment and will include these results in the camera-ready version.
>
>
> > * The paper simulates unlearning by fine-tuning a clean model on a corrupted dataset. In real-world scenarios, large language models are trained on vast corpora that may include sensitive or erroneous data. It remains unclear whether insights from these toy fine-tuning scenarios will fully translate to real LLM pretraining pipelines
> > * Finally, the paper should clarify whether and how the insights gained in the current experimental setting would generalize to full-scale pretraining scenarios.
>
> We appreciate your comment and agree that real-world scenarios are more complex. We construct a synthetic simulated task that can serve as a sanity check for unlearning methods to understand how applicable they might be in the real world. This task serves as a clean testbed for unlearning— success in our setup is likely a necessary but not sufficient condition for models to effectively have these capabilities in the real world. As we demonstrate, many prior unlearning algorithms do not have restorative unlearning capabilities even on our simplified testbed— indicating we are actually unlikely to be able to apply them effectively in more complex, real-world, scenarios.
>
>  Our simulation is constructed in the following way: we construct corruption datasets by (1) gathering information about real-world entities and (2) incorporating both incorrect and correct facts into these documents to make the corrupted documents less suspicious, and more aligned with what could happen in real-world scenarios. Our experiments were conducted with datasets containing approximately 3,000 samples and around 1,000 question/answer pairs. Furthermore, as seen in prior unlearning literature where datasets are proposed [1], model corruption is typically introduced by fine-tuning on specific documents, and unlearning is then applied to these or similarly structured documents. Thus, our approach aligns with established methodologies in the field.
>
> Note that prior benchmarks for machine unlearning [1, 2] also explore structured documents to be forgotten and typically focus on scales similar to ours for evaluation, as it enables efficient assessment. Future research could explore larger-scale experiments on restorative unlearning. We will ensure to include a discussion of larger-scale experiments in the camera-ready version of the draft.

---

> > ### Author Response · Authors · 2025-03-27
> >
> > > The paper should test different numbers of fine-tuning epochs …. and hyperparameters of fine-tuning
> >
> > Please see above response about how we considered different corruption scenarios in our existing drafts, and additional experiments over the rebuttal regarding your concern.
> >
> > [1] Maini, Pratyush, et al. "Tofu: A task of fictitious unlearning for llms." arXiv preprint arXiv:2401.06121 (2024).
> >
> > [2] Jin, Zhuoran, et al. "Rwku: Benchmarking real-world knowledge unlearning for large language models." arXiv preprint arXiv:2406.10890 (2024).

---

> > > ### Author Response · Authors · 2025-04-03
> > >
> > > We'd like to follow up to see if you have any other questions or concerns about our response that we could address before the discussion period closes. Thank you for engaging with this work. Please feel free to let us know if there is anything else you would like us to clarify.

---

> > ### Author Response · Authors · 2025-03-31
> >
> > We hope our rebuttal addressed your concerns. As the rebuttal period nears its end, we’re happy to address any final concerns.

---

### Review · Reviewer_vgTg · 2025-03-24

**Summary Of Contributions:**

This paper presents a new perspective on the evaluation of unlearning for LLM.
In particular, the authors verify whether the unlearned model can correctly respond to contaminated and uncontaminated entities by constructing a corruption dataset to simulate the scenario of a toxic data attack.

**Audience:**

Yes

**Claims And Evidence:**

Yes

**Requested Changes:**

Adding discussions of the two questions in Weaknesses may make the scenario in which the method works clearer.

**Strengths And Weaknesses:**

Strengths:
1. This evaluation method's motivation is reasonable, and the experiments designed are consistent with its basic idea.
2. The authors provide the method for constructing test data.

Weaknesses:
1. The method appears to be used in an attack scenario from design to implementation. It may not be used in unlearning scenarios that aim to protect user privacy.
2. From the experimental setup, the target model in this evaluation method is incrementally trained with toxic data in the attack phase (Corruption). Does such a setup adequately reflect the performance of unlearning methods on an LLM fully completed by extensive pre-training?

---

> ### Author Response · Authors · 2025-03-28
>
> We appreciate the reviewer’s feedback and that they find the motivation interesting, and the experiments justified..
>
> > The method appears to be used in an attack scenario from design to implementation. It may not be used in unlearning scenarios that aim to protect user privacy.
>
> Note that privacy applications for unlearning could benefit from the setting we study. A common requirement in the privacy context is to delete a datapoint that contains sensitive information. However, there is currently no consensus on what it means to ‘delete’ a data point for a model, and how we can recognize when a model has successfully done so. This is exactly the problem we aim to address— by proposing an evaluation that assesses a model's knowledge state before encountering the data, after encountering the data, and after unlearning. Ideal unlearning algorithms, even in the context of privacy concerns, should restore the model to a state as if it had never encountered the sensitive information in the first place.
>
>
>
> > From the experimental setup, the target model in this evaluation method is incrementally trained with toxic data in the attack phase (Corruption). Does such a setup adequately reflect the performance of unlearning methods on an LLM fully completed by extensive pre-training?
>
> Our setup is a simplified and more controlled version of the complexity of unlearning over extensive pretraining data— we believe that this serves as a required sanity check for unlearning procedures. If a unlearning algorithm is to be effective on pretraining data, it should be effective on our simplified and more minimal tests. Notably, we observe that prominent methods struggle even in this controlled setting, raising concerns about their ability to perform effective unlearning in real pretraining corpora.
>
>
> We designed a scenario where knowledge corruption occurs in a continual pretraining setup, and unlearning must restore the model to its knowledge state before encountering the unlearning target, i.e., not only remove the corrupted knowledge but also recover the correct knowledge present in other documents the model has encountered. Notably, this methodology aligns with existing benchmarks like TOFU [1], where models are incrementally trained on a dataset that is later used for unlearning. Finally, similar to other works in the literature, the effectiveness of these unlearning methods in our scenarios is likely a necessary but not sufficient condition for their success in real-world applications, given that our setup is a simplified simulation.
>
>
> [1] Maini, Pratyush, et al. "Tofu: A task of fictitious unlearning for llms." arXiv preprint arXiv:2401.06121 (2024).

---

> > ### Author Response · Authors · 2025-03-31
> >
> > We hope our rebuttal has addressed your comments. As the rebuttal period nears its end, we’re happy to clarify any remaining concerns.

---

> > ### Author Response · Authors · 2025-04-03
> >
> > We'd like to follow up to see if you have any other questions or concerns about our response that we could address before the discussion period closes. Thank you for engaging with this work. Please feel free to let us know if there is anything else you would like us to clarify.

---

### Review · Reviewer_SQyr · 2025-03-27

**Summary Of Contributions:**

RESTOR introduces a framework for evaluating machine unlearning that restores a model’s original knowledge by erasing corrupted training data. The study compares methods, including Gradient Ascent, KL Divergence, and Negative Preference Optimization, and then finds that preference-based unlearning more effectively recovers factual accuracy.

**Audience:**

Yes

**Claims And Evidence:**

Yes

**Requested Changes:**

Please refer to the weaknesses.

**Strengths And Weaknesses:**

Strengths:

1. The evaluation proposed by this paper is interesting and reasonable.

2. The structure of this paper is clear.

Weaknesses:

1. The differences between RESTORE and the existing poison data unlearning are not explained. If the corrupted data exists in the training dataset, there will be no difference between the two evaluation scenarios.
2. Another concern is whether the corruption step can affect the other knowledge beyond the data in $\mathcal{F}$. If so, the recovery of the related knowledge should also be examined.
3. The synthetic data limited the generalizations of the experiments. More synthetic data on different topics can be more convincing.

---

> ### Author Response · Authors · 2025-03-28
>
> We appreciate that the reviewer finds our evaluation interesting, and our paper well-written. Regarding your concerns:
>
> > The differences between RESTOR and the existing poison data unlearning are not explained. If the corrupted data exists in the training dataset, there will be no difference between the two evaluation scenarios.
>
> Thanks for raising this point. Note that in this work, we focused on this problem in the context of large language models. To the best of our knowledge, evaluating machine unlearning in large language models for restoration—where data poisoning can be seen as one application—has not been explored in the literature. We will ensure that related work on unlearning for data poisoning is included in the camera-ready version. Existing research on this topic [1, 2, 3] has focused on the vision domain rather than the language model space.
>
>
> > Another concern is whether the corruption step can affect the other knowledge beyond the data in $\mathcal{F}$. If so, the recovery of the related knowledge should also be examined.
>
> Thank you for the excellent suggestion. We will add the performance on untargeted entities to the final version, which are an example of knowledge that was not the target in corruption or unlearning. Corrupted datapoints do indeed have second-order effects on entities beyond the targeted ones, and we observe that algorithms that successfully restore performance on the targeted entities, also successfully undo this second-order corruption. We will add the results of these experiments  to the draft.
>
> In the literature (existing benchmarks), studies on the side effects of unlearning algorithms have mainly focused on model utility, as these algorithms are generally effective at forgetting (achieving high forgetting scores) but often at the cost of substantial utility degradation. As a result, prior work has primarily investigated this trade-off by evaluating unlearned models on related concepts. We shed light on a complementary direction to these prior studies: restoring the model’s original knowledge state, rather than focusing on the general notion of “utility” which has been studied in prior work.
>
> > The synthetic data limited the generalizations of the experiments. More synthetic data on different topics can be more convincing.
>
> Thanks for raising this point. We agree that more research can be done on restorative unlearning from the data perspective. However, it’s important to note that our work is the first to propose this task for unlearning in language models, providing a benchmark and a systematic evaluation of various unlearning algorithms for the task. We demonstrated that, even in our simple scenario, many algorithms fail significantly, which shows that they are not capable of restorative unlearning, even in relatively straightforward cases.
>
> Furthermore, as discussed at the end of Section 4.3 (Broad-spectrum model corruption), we encountered corruption scenarios where none of the algorithms were effective, underscoring the difficulty of the problem—even for NPO, which performed well in simpler cases. These results highlight the complexity of the task and emphasize the need for future developments in unlearning algorithms.
>
> To summarize, our primary focus was to build a controlled simulation for evaluating unlearning,  where we demonstrated the ineffectiveness of multiple prominent unlearning algorithms. We also extended the study further to explore more severe forms of knowledge degradation, where restorative unlearning is made challenging. This aligns with your suggestion that data plays a crucial role in this task, and it opens up additional research directions for future work in this area.
>
> -----------
>
> [1] Pawelczyk, Martin, et al. "Machine unlearning fails to remove data poisoning attacks." arXiv preprint arXiv:2406.17216(2024).
>
> [2] Li, Wenjie, et al. "Delta-Influence: Unlearning Poisons via Influence Functions." arXiv preprint arXiv:2411.13731 (2024).
>
> [3] Schoepf, Stefan, Jack Foster, and Alexandra Brintrup. "Potion: Towards poison unlearning." arXiv preprint arXiv:2406.09173(2024).

---

> > ### Author Response · Authors · 2025-03-31
> >
> > As we approach the end of the rebuttal period, we are happy to address any remaining concerns. Let us know if your concerns have been resolved.

---

> > ### Author Response · Authors · 2025-04-03
> >
> > We want to follow up to see if you have any additional questions or concerns about our response that we could address before the discussion period closes. Thank you for engaging with this work. Please feel free to let us know if there is anything else you would like us to clarify.

---

### Comment · Action_Editor_seFJ · 2025-03-20
**Rebuttal**

Dear authors,

We have collected three reviews. You have two weeks to do the rebuttal.

Best wishes,
Tongliang

---

> ### Author Response · Authors · 2025-03-28
>
> Dear Prof. Liu,
>
> Thank you for your message and coordination of the review process. We appreciate valuable feedback from the reviewers and have carefully addressed their comments in our rebuttal. We have submitted our response.
>
> We look forward to the next steps of the process.
>
>
>
> Best, Authors.

---

### Decision · Action_Editor_seFJ · 2025-05-03

**Recommendation:** Accept with minor revision

**Comment:**

This paper introduces a framework to evaluate machine unlearning in LLMs through the lens of knowledge recovery rather than forgetting alone.

Reviewers commonly agree that the paper presents a novel perspective on restorative unlearning (Reviewer kKUp), features high-quality and well-written presentation (Reviewer BuHh, Reviewer ExCS), and offers insightful empirical analysis on the limitations of existing unlearning methods (Reviewer BuHh).

There are some remaining concerns, such as limited evaluation on real-scale LLMs (Reviewer vgTg, Reviewer ExCS) and a lack of discussion on privacy-oriented unlearning settings (Reviewer vgTg). These concerns which do not affect the main claims of the paper are considered minor and can be addressed in a minor revision or left as future work. Therefore, the AC recommends accepting this paper.

The authors need to clearly revise the submission to incorporate clarifications raised during the review, especially regarding the scope of applicability, connections to related work in privacy and corrective unlearning, the implications for scaling the proposed framework to real-world LLM deployments, and limitations.

**Audience:**

Understanding how to remove or recover specific knowledge from trained models is an important problem in trustworthy and responsible AI. The topic of this paper is of clear interest to researchers working on unlearning in LLMs.

**Claims And Evidence:**

Yes, major claims are well supported by clear empirical analysis, systematic experimental design, and comparisons with existing unlearning baselines.

---

> ### Author Response · Authors · 2025-05-11
>
> Thank you for the thoughtful evaluation and for accepting our paper!
>
> We’re grateful for the reviewers’ insights and appreciative of the opportunity to share our work. In the final version, we will incorporate the suggested clarifications, including connection to related work, scope of applicability, and the scalability of our framework.
>
>
> We plan to submit the camera-ready version within the next two weeks.